# Participation in Activities Fostering Children’s Development and Parental Concerns about Children’s Development: Results from a Population-Health Survey of Children Aged 0–5 Years in Quebec, Canada

**DOI:** 10.3390/ijerph17082878

**Published:** 2020-04-21

**Authors:** Gabrielle Pratte, Mélanie Couture, Marie-Ève Boisvert, Irma Clapperton, Josiane Bergeron, Marie-Andrée Roy, Élyse Dion, Chantal Camden

**Affiliations:** 1Faculté de Médecine et des Sciences de la Santé, Université de Sherbrooke, Sherbrooke, QC J1K 2R1, Canada; gabrielle.pratte@usherbrooke.ca (G.P.); marie-eve.boisvert.5@ulaval.ca (M.-È.B.); irma.clapperton.ciussse-chus@ssss.gouv.qc.ca (I.C.); chantal.camden@usherbrooke.ca (C.C.); 2Institut Universitaire de Première Ligne en Santé et Services Sociaux (IUPLSSS), Sherbrooke, QC J1H 4C4, Canada; 3Institut Universitaire en Déficience Intellectuelle et Trouble du Spectre de l’autisme, Trois-Rivières, QC G8Z 3T1, Canada; 4Centre de Recherche du Centre Hospitalier Universitaire de Sherbrooke (CR-CHUS), Sherbrooke, QC J1H 5N4, Canada; 5Direction de la Santé Publique de l’Estrie, Sherbrooke, QC J1J 3H5, Canada; 6Projet Partenaires pour la Réussite Éducative en Estrie (Projet PRÉE), Magog, QC J1X 3H2, Canada; bergeronj@etsb.qc.ca (J.B.); elyse.dion@csdessommets.qc.ca (É.D.); 7Collectif Estrien 0-5 ans, Sherbrooke, QC J1H 4C4, Canada; chus@ssss.gouv.qc.ca

**Keywords:** child development, population characteristics, public health, reading, motor skills

## Abstract

This study aims to: (1) describe children’s participation in activities fostering their development, (2) document parental concerns about their children’s development, and (3) explore the influences of family characteristics on children’s activity participation and parental concerns. We conducted a phone survey with parents of children aged 0-5 years (*n* = 895). Survey results are presented as weighted proportions for the parent’s age, sex, and area of residence. Statistical comparisons were made using chi-square with *p* < 0.05. Most children were exposed at least weekly to fine motor (85.1% ± 2.4%), physical (83.0% ± 2.5%), and reading (84.2% ± 2.4%) activities. However, only a small proportion were exposed to those activities daily (49.7% ± 3.3%, 35.4% ± 3.2%, and 32.4% ± 3.1% respectively). Many (46.8%) parents had concerns about their children’s development. The most frequent domains of concern were communication skills (22.8% ± 2.8%), affective and behaviour skills (22.1% ± 2.7%), and autonomy (19.6% ± 2.6%). The proportion of parents having concerns was higher among families with lower incomes. The small proportion of children exposed daily to activities fostering their development, and the high proportion of parents with concerns about their children’s development are alarming. The integration of health and education services and the use of best practices fostering children’s development at home, at school, and in daycare centres is needed.

## 1. Introduction

Child development is influenced by social circumstances in early life [1,2], and poor development has lifelong effects on literacy, numeracy, school achievement, subjective well-being, and behaviour [3,4,5,6]. Given the importance of early childhood development, it is not surprising that many communities have established programs to monitor children’s development, and to quickly identify and refer to early intervention those children at risk of developmental delays. At the public health level, these programs should also provide helpful information on children’s developmental issues, and support the advancement of interventions to promote optimal child development [7]. 

In Quebec, several studies have employed a population approach to monitor children’s development. For instance, the Quebec Survey of Child Development in Kindergarten (EQDEM) was a provincial school readiness and global development study of children conducted in 2012 and 2017 [8,9] using the early development instrument (EDI) [10]. The EDI was also used in the Quebec Longitudinal Study of Child Development (ELDEQ) [11]. The EDI is widely used to measure children’s global development from the kindergarten teacher’s perspective. The EDI measures child development in five domains: physical health and well-being (i.e., motor skills, cleanliness, and adequate clothing), social competency (i.e., respect of rules, peers, and routine), affective maturity (i.e., fears, aggressive behaviour, and expression of emotions), cognitive and language development (i.e., reading and mathematics skills), and communication skills and general knowledge (i.e., understanding and being understood, and general knowledge) [10]. According to the EQDEM results, 27.7% of kindergarten children in Quebec are vulnerable in at least one developmental domain. In the region of Estrie, Quebec, this proportion is slightly higher, at 29.4% [12], and of greatest concern, has increased significantly in four out of the five developmental domains between 2012 and 2017 despite regional actions to support children’s development [9]. The Estrie is also a region characterized by a high rate of children living in families with low income and living in a family with no parents having at least a high school degree in comparison with other regions in the province of Quebec [12,13].

A group of stakeholders concerned about early childhood development in Estrie decided to work together to monitor and further support children’s development. Since recent provincial data provide primarily teachers’ perspectives about children’s development and school readiness [9,14], and given the importance of the parental perspective in the development of early childhood intervention [15], the purpose of this study was to survey parents in Estrie about their children’s development. 

In accordance with governmental guidelines, we chose a global development perspective to describe pre-schoolers’ development, rather than a school readiness perspective, which is sometimes understood as limited to cognitive and language skills or specific knowledge [16,17]. To do so, we used a framework increasingly used in Quebec, and among health professionals, including five developmental domains: affective (e.g., managing emotions and self-confidence), cognitive (e.g., learning and memorization), social (e.g., making friends), language (e.g., speaking and understanding), and physical and motor development (e.g., run and manipulate objects) [16]. We were specifically interested in exploring children’s participation in activities contributing to their development in these developmental domains, and their parents’ developmental concerns. Specific aims of this study were to: (1) describe children’s participation in activities fostering their development, (2) document parents’ developmental concerns, and (3) explore the influence of family characteristics on children’s activity participation and parental concerns. 

## 2. Materials and Methods 

### 2.1. Design

As part of the Estrie Population Health Survey—Enquête de santé publique de l’Estrie (ESPE), questions pertaining to children’s development were proposed. The ESPE is a regional survey that was last conducted in 2018 by the regional public health services, in collaboration with several researchers from different fields. The ESPE aims to monitor population health and well-being and describe health determinants (e.g., tobacco amd obesity) to support the development of a regional action plan. Regional stakeholders involved in child development initiatives proposed survey questions that were submitted to a central committee responsible for the whole survey, which made the final selection. This paper focuses on results emerging from those questions. 

### 2.2. Sample 

Participants were adults aged 18 years and older, living in Estrie. A random sample of 46,356 land-line and mobile phone numbers were selected based on the nine local service networks, to ensure a representative sample of the regional population. Among valid phone numbers, a response rate of 40% was determined for a total desired sample of 10,650 respondents. This response rate was fixed due to survey length and the moderate to high importance of the survey, suggesting a target response rate between 20% and 40% [18]. The subsample of participants included in this paper was restricted to ESPE respondents who reported living with at least one child between 0 and 5 years of age in the sociodemographic questions of the survey and who completed the additional questions about their children’s development. These questions were asked at the end of the survey. If the participant had more than one child of this age, the participant was asked to answer for the oldest child. Either the father or the mother of this child could respond to these questions, depending who was completing the survey.

### 2.3. Data Collection

Data were collected via phone with an online completion option offered to respondents who preferred online completion or were not available by phone. The survey was available in French and English. The average duration to complete all questions of the ESPE survey was 34 min. The survey included socio-demographic data, such as respondent age, sex, socio-economic status, and territorial and local service networks. The parent survey also included questions about: (1) their children’s participation in activities fostering child development, and (2) parental concerns about their children’s development. 

Survey questions about children’s activity participation inquired about reading activities (“During the past 12 months, how often did you and your child, or any other adult in the household with your child, do reading activities (e.g., reading a book or telling a story, going to the library or bookstore)?”), fine motor activities (“During the past 12 months, how often did your child engage in fine motor skills such as tinkering, drawing, gluing or cutting, or activities such as performing arts, painting, sculpture?”), and physical activities (“During the past 12 months, how often did your child participate in physical activity or free sport with or without a coach or instructor (e.g., karate class, playing ball, jumping rope, riding a bike, go swimming)?”) with 7 response options (“rarely or never,” “less than once a month,” “once a month,” “a few times a month,” “once per week,” and “a few times a week”). Parents were also asked: “How often do you visit places related to reading (library, bookstore...)?” and “Does your child participate in the following activities or programs? (a) Community class or activity for parents and children? (b) Story time in a library or other reading program or club?” Those questions were based on the ELDEQ [19] and the Progress in International Reading Literacy (PIRLS) [20] surveys. 

For parental concerns, each of the five developmental domains (i.e., affective, cognitive, social, language, and motor development) [16] were included (e.g., “Do you have, or have you ever had, concerns about the development of your child, specifically regarding: his/her motor skills (running, balancing, taking, and handling small objects)?”). Response options for these questions were yes and no. For parents responding yes to at least one developmental domain, they were asked if they had sought support from rehabilitation or health professionals. Response options included “I consulted and it has met our needs,” “I consulted and this did not meet our needs,” “I plan to consult soon,” “I do not need to consult,” and “I am waiting for services.” 

### 2.4. Data Analysis

Data were collected and analysed in Field Track software by interviewers hired from a professional firm. Descriptive statistics were calculated for all data, but percentages reported in Section 3.2 and Section 3.3 were weighted for age, sex, and area of residence to better characterize Estrie’s population. To ease data reporting and comparison, data were clustered as appropriate (e.g., response options were dichotomized in ‘’at least once a week” and “less than once a week” for frequency of children’s participation in activities). Statistical differences reported in the tables refer to differences between a category and the rest of the sample using 2 by 2 matrices and chi-square tests. Statistical significance was set at *p* < 0.05. 

## 3. Results

### 3.1. Participant Characteristics

Between June 18 and November 12 2018, 9267 surveys were conducted by phone and 1523 were completed online for a total of 10,790 participants (for an ESPE survey response rate of 40.1%). Participants had an average age of 35.5 years and 1.43 children per family (see Table 1). Among respondents of the ESPE survey, 11.5% (*n* = 1,240) had a least one child between 0 and 5 years of age in their household. Of those, 72.2% (*n* = 895) completed the questions related to their children’s development. 

### 3.2. Children’s Activity Participation 

More than 80% of parents reported that their children completed fine motor (85.1% ± 2.4%), physical (83.0% ± 2.5%), and reading (84.2% ± 2.4%) activities at least once a week (Table 2). Percentages of families reporting that their children engaged in those activities daily were significantly lower (49.7% ± 3.3% for fine motor; 35.4% ± 3.2% for physical, and 32.4% ± 3.1% for reading activities). Family participation in library story time or community classes/activities for parents and children was significantly lower among parents aged 18 to 29 years. A trend was also observed for reading habits with the older parents more likely to visit places related to reading with their children when compared with younger parents.

### 3.3. Parents’ Developmental Concerns 

Parent domains of concern were communication skills (22.8% ± 2.8%), affective and behavioural skills (22.1% ± 2.7%), and autonomy (19.6% ± 2.6%) (Table 3). Families with lower incomes had significantly higher percentages of concerns in all domains except for the social skills domain. Younger parents (18–29 years) were less concerned than older parents for the behavioural, social, and cognitive domains. Fathers had significantly fewer concerns than mothers in the communication domain only. 

Among parents with concerns in at least one developmental domain (*n* = 419, 46.8%), 52.7% consulted rehabilitation or health professionals: 2.9 ±1.6% responded “I consulted and this did not meet our needs,” 13.5 ± 3.3% “I consulted and it has met our needs,” and 36.3 ± 4.6% “I consulted and it met our needs very well.” Among families who did not consult, 34.1 ± 4.5% considered they did not need services despite their concerns, and 8.3 ± 2.6% were waiting for services. 

## 4. Discussion

We sought to describe children’s participation in activities fostering development, and to document parental concerns about their children’s development. Results highlighted the low percentage of families reporting their children’s daily engagement in activities contributing to their development, and the high percentage of families reporting developmental concerns. The discussion is organized around these two mains findings, and integrated with our findings regarding the influence of family characteristics on children’s participation and parental concerns. 

### 4.1. Children’s Activity Participation 

Results revealed that most children engaged in fine motor, physical, and reading activities at least once a week. However, 14.9%–17.0% of parents reported that their children engaged in those activities only a few times a month or less. This is a concern because those activities are known to be important for child development and are often recommended on a daily basis to develop important skills and habits for school [19,21]. 

For instance, reading with parents once a month or less is associated with higher proportion of vulnerability in at least one developmental domain, according to the EQDEM [14]. Similarly, results of studies using the ELDEQ found that reading with young children is associated with higher receptive language at three years of age [22], and that not reading daily with children aged 18 months is associated with higher percentages of vulnerability in at least one developmental domain in kindergarten [19]. Our findings on parental reading habits (32.4% for daily reading and 56.5% for library visits) are lower than those reported using the EQPPEM (*Enquête Québécoise sur le parcours préscolaire des enfants à la maternelle*) where 41% of children read daily with their parents, and 68.5% of parents visited the library at least occasionally with their children [14]. We found that reading habits did not seem to be associated with familial income, even if socio-economic status is frequently associated with a higher proportion of children at risk of developmental delay [14]. A recent provincial portrait of parenting behaviours revealed that an important factor influencing parental reading habits is parental sense of competency [23]. Public health initiatives fostering parental sense of competency with regard to reading skills for all families, independent of their socio-economic characteristics, may be an important strategy to foster optimal child development. 

Child participation in fine motor and physical activities is less documented in the literature. Nevertheless, it is worrisome that less than half of children engaged in daily physical activities, with only approximately 35% exposed to daily fine motor activities. Guidelines recommend at least 180 minutes of physical activity for children aged 1–4 years, and at least 30 minutes of daily “tummy time” for infants not yet mobile [21]. Our results may, however, need to be interpreted with caution, as parents may not be aware of the activities considered as fine motor or physical activities, especially for children younger than 1 year (e.g., tummy time could have been considered as motor activities, or not, depending on parental perceptions, knowledge, and expectations related to physical activity). An alternate hypothesis is that our findings do reflect accurate fine motor and physical activity times, with the results explained by the increasing screen time seen among young children. Considering the importance of parental influence on child sedentary behavior at home [24], and the importance of daycare centre time among children in Quebec [14], efforts to promote physical activities, and the provision of opportunities to develop fine and global motor skills by all stakeholders involved with children are critical [7]. 

### 4.2. Parental Concerns 

The percentage of parents reporting developmental concerns (46.8%) is consistent with a recent local study that found that 46.2% of parents had concerns about their children’s development [25]. Considering the representativeness of our sample, the proportion of parents concerned with their children’s development is alarming, particularly since another regional study showed that the percentages of parents, who reported child development concerns were similar to the proportion of children at risk of delay in at least one domain of the Ages and Stages Questionnaire (ASQ-3) (44.8%) [25]. Similarly, Chung and colleagues [26] found that parents are particularly good at detecting communication and motor difficulties, and when they do consult a professional, a diagnosis is often given, confirming a delay [26]. Those results support the need for a continuum of services, including various strategies to promote children’s development, and rehabilitation, as needed. For some parents, it may also highlight the need to be reassured about their children’s development, which is, in itself, an important health promotion and self-efficacy action.

Our results are complementary to developmental vulnerability observed by kindergarten teachers when using the EDI [9,12]. The percentage of parents having concerns in at least one developmental domain (46.8%) may seem high compared to vulnerability measured by teachers on the EDI (29.4%) [12]. However, those results are not contradictory and should be used for different purposes because they: (1) represent the perceptions of different stakeholders; (2) use different definitions of a child considered to have vulnerable development; and (3) cover different developmental domains.

When seeking a complete portrait of children’s needs at a population level, it is important to include various stakeholders, such as parents, teachers, and the community because each has different points of view [15]. For instance, the children who are vulnerable with regard to their development, as measured by teachers, are interesting to compare different populations (Moisan 2014). However, when the goal is to develop actions to foster children’s development, parental perceptions should also be considered, given the parents’ role as the persons ultimately responsible for their children’s development [15]. 

When the goal is to determine actions to foster child development, a population portrait based on clinical thresholds or age-based expectations (such as parental concerns) could be more insightful than a relative measure of child development. An example of the relative concept of vulnerability is the EDI measure, which postulates that children are vulnerable if they are part of the 10% having the lowest score in a specific domain [17]. This relative concept of vulnerability is useful to determine if a specific population is more or less vulnerable than another [17]. However, parental concerns, based on expectations related to the child’s age, and lived challenges could provide more important insights when developing local initiatives aimed at fostering child development [15]. 

The domains of greatest concern for parents in our study were communication skills, affective and behavioural skills, and autonomy. Those domains differ slightly from the principal domains of vulnerability in the EQDEM in Estrie, which were cognitive and language (i.e., reading and writing skills), affective maturity, and social competency [9]. It is interesting to note that the use of developmental domains suggested by the Ministry of Family [16] for our survey questions, instead of EDI domains, allowed us to discover that a large proportion of parents have concerns about their children’s autonomy in their daily activities (19.6%) and motor skills (13.7%). This proportion of parents may have been underestimated by the EDI because autonomy and motor skills are part of the wide category of physical health and well-being [10,16]. Another regional survey found that the domain in which children are the most often at risk of developmental delay is fine motor skills [25]. Our results demonstrate a clear difference between percentages of parental concerns regarding communication skills (i.e., understanding and speaking)—which was the first source of concern for parents in this survey (22.6%)—and cognitive skills (9.4%). This difference cannot be detected by the EDI because the domain “communication skills and general knowledge” includes abilities to understand and be understood, which refers to our definition of communication skills, and cognitive elements such as general knowledge [9]. 

Our results suggest that families with low incomes have more developmental concerns for their children than those with higher incomes. This contrasts with the widely accepted view that disadvantaged families are less aware of their children’s difficulties [27]. However, these results are consistent with the fact that children living in low income families are more at risk of developmental delay [9,25]. The finding of a similar level of concern between mothers and fathers with the exception of the communication domain is surprising, as mothers often seem more worried about their children’s development and health than fathers [28].

Our finding that roughly one third of parents who have concerns do not feel the need to consult rehabilitation or health professionals is surprising. However, the processes of recognizing developmental difficulties and the decision-making to consult health or rehabilitation professionals are separate processes. Help-seeking behaviors can be influenced by many factors; for example, type and amount of delay, cultural expectations, perceptions of children’s development, and satisfaction with services [29,30]. Interestingly, among families in our study who consulted health professionals related to their concerns, the proportion of families dissatisfied with services received (2.9%) was relatively low compared to provincial statistics estimating the proportion of dissatisfaction with public health services at 9% [31].

### 4.3. Strengths and Limitations

Given its population focus, our study was limited by the specific developmental topics covered by the survey. Several variables that may have being helpful for our analysis, such as child age, attendance at childcare center, and parental education were not available. Further study should be conducted to explore effect of these variables on parental concerns and children participation in activities. While the 40.1% response rate may seem low, it represents the upper bracket of the Canadian response rate target for a survey of this importance [18]. Despite random sampling, selection bias might have occurred at different stages of the study, and we have no information about why some families having children 0–5 years old did not respond to the questions about their children’s development. It is possible that some subgroups of the population are underrepresented in this study, but all possible measures have been taken to limit bias (e.g., survey available in French and English, to be completed online or on the phone when participants were available). The final sample was quite representative of the target population on all socio-demographic factors except for families speaking a language other than English or French. The representativeness and the overall sample size are clearly strengths of our study. Strategies we used to reach young adults, who are frequently harder to reach in phone surveys, partly because of the use of mobile phones, were effective in ensuring the representation of young parents. These results are specific to the particular region of Estrie, but might be similar in other regions, especially those with lower socio-economic status.

## 5. Conclusions

Our results described a high rate of developmental concerns in Estrie, Quebec. Developmental domains of greatest concern were communication skills, affective and behavioural skills, and autonomy, which differ from traditional domains of vulnerability identified in school-based surveys exploring teachers’ perceptions. Differences in stakeholders’ perceptions highlight the need to take a broad perspective of children’s development, and call for greater integration of education and health care services. It is also essential to encourage parents to voice their developmental concerns, and to provide appropriate services, and/or support/reassurance as necessary. Most importantly, the small proportion of children exposed to daily motor and/or reading activities calls for the integration of health promotion best practices aimed at fostering children’s development at home, at school, and in daycare centres. Finally, efforts to develop a common language around child development that includes all aspects of child development, including those that are important for children, parents, and other stakeholders to foster cost-effective, global support to child development are critically needed. 

## Figures and Tables

**Table 1 ijerph-17-02878-t001:** Socio-demographic characteristics (n = 895).

Socio-Demographic Characteristics	Number of Families (*n*)	Proportion (%)
Number of children in the family		
1 child	556	61%
2 children	300	34%
≥ 3 children	39	4%
Parent’s age (years)		
18–29	127	14%
30–44	714	80%
45–64	51	6%
>64	3	< 1%
Composition of the household		
Couple	771	86%
Single-parent family	81	9%
Other	39	4%
Household income		
< $30,000	80	9%
$30,000–$79,999	380	42%
> $80,000	401	45%
Language spoken at home		
French	826	92%
English	59	7%
Other	25	3%

**Table 2 ijerph-17-02878-t002:** Proportion of parents reporting child’s activity participation (%, ± 95% CI).

Child’s Activity Participation	Total Proportion	<$30,000 Income	$30,000–$79,999 Income	>$80,000 Income	Parent 18–29 Years-Old	Parent 30–44 Years-Old	Parent 4 45–64 Years-Old
Fine motor activities ^a^	85.1 ± 2.4%	87.5 ± 7.3%	84.9 ± 3.7%	85.7 ± 3.4%	76.5 ± 7.8%	87.7 ± 2.4%	92.6 ± 7.5%
Physical activities ^a^	83.0 ± 2.5%	87.6 ± 7.3%	84.7 ± 3.7%	82.5 ± 3.7%	70.8 ± 8.2%	87.3 ± 2.5%	84.3 ± 10.3%
Reading activities ^a^	84.2 ± 2.4%	82.4 ± 8.4%	81.5 ± 3.9%	88.7 ± 3.1%	68.7 ± 8.2%	89.8 ± 2.2%	89.4 ± 8.9%
Visit places related to reading ^b^	56.5 ± 3.3%	10.2 ± 6.7%	14.7 ± 3.6%	13.6 ± 3.4%	10.3* ± 5.3%	14.1 ± 2.6%	28.1* ± 12.7%
Participate in library story time	24.8 ± 2.8%	32.3 ± 10.4%	24.1 ± 4.3%	24.4 ± 4.2%	16.6* ± 6.6%	27.3* ± 3.3%	32.2 ± 13.1%
Participate in community class or activity for parents and children	28.5 ± 3.0%	32.3 ± 10.4%	26.6% ± 4.5%	30.2 ± 4.5%	19.2* ± 7.0%	32.1* ± 3.4%	26.1 ± 12.6%

^a^ At least once a week; ^b^ each week or occasionally. * A statistically significant difference between the proportion in a specific category when compared to the proportion among all other participants, at *p* < 0.05

**Table 3 ijerph-17-02878-t003:** Proportion of parents reporting concerns in each developmental domain (%, ± 95% CI).

Developmental Domains	Total Proportion	< $30,000 Income	$30,000–$79,999 Income	> $80,000 Income	Parent 18–29 Years-Old	Parent 30–44 Years-Old	Parent 45–64 Years-Old	Mother	Father
Communication skills	22.8 ± 2.8%	32.3 * ± 10.4%	24.9 ± 4.4%	19.5* ± 3.9%	22.2 ± 7.4%	23.5 ± 3.1%	16.1 ± 10.5%	25.8 * ± 3.6%	19.3 * ± 4.5%
Affective and behaviour skills	22.1 ± 2.7%	34.7 * ± 10.4%	20.7 ± 4.1%	22.7 ± 4.1%	15.0 * ± 6.2%	25.1 * ± 3.2%	18.7 ± 10.7%	23.5 ± 3.4%	20.4 ± 4.5%
Child autonomy	19.6 ± 2.6%	27.0 * ± 10.0%	17.6 ± 3.9%	20.0 ± 3.9%	18.3 ± 6.9%	20.5 ± 3.0%	11.5 ± 9.0%	19.9 ± 3.3%	19.2 ± 4.4%
Social skills	17.0 ± 2.5%	24.2 ± 9.6%	19.1 ± 4.0%	14.4* ± 3.5%	11.0 * ± 5.6%	19.2 * ± 2.9%	18.1 ± 11.0%	17.7 ± 3.1%	16.2 ± 4.2%
Motor skills	13.7 ± 2.3%	24.8 * ± 9.7%	12.3 ± 3.3%	12.5 ± 3.2%	12.1 ± 5.8%	14.5 ± 2.6%	9.6 ± 8.3%	13.5 ± 2.8%	13.9 ± 3.9%
Cognitive skills	9.4 ± 2.0%	18.5 * ± 8.7%	9.7 ± 3.0%	7.6* ± 2.6%	6.4 * ± 4.4%	10.0 ± 2.2%	14.2 ± 10.0%	9.1 ± 2.3%	9.5 ± 3.3%

* A statistically significant difference between the proportion in a specific category when compared to the proportion among all other participants, at *p* < 0.05.

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
