# Peer review of "Participation in Activities Fostering Children’s Development and Parental Concerns about Children’s Development: Results from a Population-Health Survey of Children Aged 0–5 Years in Quebec, Canada"

_ijerph, 2020, doi:10.3390/ijerph17082878_

Round 1

Reviewer 1 Report

I suggest to expand information about capture Method how many by phone? how many by online? survey completaron?

What do you think about selection bias? Could change your results?

Author Response

Thank you very much for reviewing our manuscript. Modifications made as per your suggestions (bold-face type), are detailed below (in italic). Modification of the original text is highlighted in red in the text.

Point 1: I suggest to expand information about capture Method how many by phone? how many by online? survey completaron?

 Response 1: Information about phone and online survey was added (lines 141-142): “Between June 18th and November 12th 2018, 9,267 surveys were completed by phone and 1,523 online, for a total of 10,790 participants (for an ESPE survey response rate of 40.1%).”

Point 2: What do you think about selection bias? Could change your results?

Response 2: Random sampling limited selection bias. However, refusal rate might have been higher for some groups, and we added it in our limitations (lines 300-308): “om sampling, selection bias might have occurred at different stages of the study, and we have no information about why some families having a child 0-5 years old did not respond to the questions about children’s development. It is possible that some subgroups of the population are underrepresented in this study, but all possible measures have been taken to limit bias (e.g. survey available in French and English, to be completed online or on the phone when participants were available). The final sample was quite representative of the target population on all socio-demographic factors except for families speaking a language other than English or French. The representativeness and the overall sample size are clearly strengths of our study.

Thank you for considering this manuscript for publication in IJERPH.

Sincerely,

Gabrielle Pratte

On behalf of the study co-authors

Reviewer 2 Report

This report of a population survey of families of developmental activities and expectations for children under 5 in Quebec raises interesting issues around family developmental activities and family concerns about young children's development.  However, there are a number of issues that should be addressed to strengthen the understanding and utility of the study.  First, what are the overall population differences in the particular sample area vs other areas of the province and how might this influence the outcomes?  Second, while the sampling process was well described, the text reports 1240 completed interviews while the tables show n=895.  Why is there a difference and is it systematic? Third, Table 3 reports mother and father differences but it is unclear if both the mother and father in each family were interviewed or if this just represents who happened to be interviewed in each family (with one interview per family).

More significant, there seem to be two major deficits in the way data were collected that present large limitations on the usefulness of the study.  One is that there appears to be no data on how many children are in child care settings where presumably many of the activities asked about do occur.  So were families asked only about home-based activities? If families had multiple children, they were asked to only describe the oldest child which means the sample might be skewed to older children who more likely would be in nursery or preschool settings?  If this isn't know are there other data about preschool enrollment/child care rates in the area that might lend some insight? The age of the child is also extremely important since for parents of infants or toddlers, some of the activities might be less relevant or harder for them to answer accurately (for example would parents interpret 'tummy time' as physical exercise/motor development?)  These deficits in study design make the findings a lot less useful. Parental expectations and knowledge would also figure into their ratings and might differ by age of child, and if they had only on or more than one children.  The latter analysis seemingly could be done.  While these issues are alluded to in the discussion, along with limitations about no age data, the authors need to expand the discussion on these points and link implications or actions to a range of possible interpretations, including possible follow up studies that obtain more in depth information from a smaller sample of families with young children. 

Author Response

Thank you very much for reviewing our manuscript. We found your suggestions and comments very helpful and have made revisions to the manuscript accordingly. Modifications made as per your suggestions (bold-face type), are detailed below (in italic). Modification of the original text is highlighted in red in the text.

Point 1: This report of a population survey of families of developmental activities and expectations for children under 5 in Quebec raises interesting issues around family developmental activities and family concerns about young children's development.  However, there are a number of issues that should be addressed to strengthen the understanding and utility of the study.  First, what are the overall population differences in the particular sample area vs other areas of the province and how might this influence the outcomes?

Response 1: We added information about socio-economic particularity of the Estrie in the introduction section: “According to the EQDEM results, 27.7% of kindergarten children in Quebec are vulnerable in at least one developmental domain. In the region of Estrie, Quebec, this proportion is slightly higher, at 29.4% [12], and of greatest concern, has increased significantly in four out of the five developmental domains between 2012 and 2017 despite regional actions to support children’s development [9]. The Estrie is also a region characterized by a high rate of children living in families with low income and living with no parents having at least a high school degree in comparison with other regions in the province of Quebec [12,13].” (lines 54-60). We also added a few words about it at the end of the Strengths and limitations section: “These results are specific to the particular region of Estrie, but might be similar in other regions, especially those with lower socio-economic status.” (lines 310-311).

Point 2: Second, while the sampling process was well described, the text reports 1240 completed interviews while the tables show n=895. Why is there a difference and is it systematic?

Response 2: 1240 represents the number of participants in the ESPE survey that declared living with at least one child between 0 and 5 years old in their household (question at the beginning of the sociodemographic section in the survey). 895 represents the number of participants that completed the child development questions. We modified the following sentences in the sample section to ensure greater clarity: “The subsample of participants included in this paper is restricted to ESPE respondents who reported living with at least one child between 0 and 5 years of age in the sociodemographic questions of the survey and who completed the additional questions about their child’s development. These questions were asked at the end of the survey”(lines 96-99). We also added the following in the results section: Among respondents of the ESPE survey, 11.5% (n=1,240) had a least one child between 0 and 5 years of age in their household. Of those, 72.2% (n=895) completed the questions related to their child development.” (lines 143-146). No specific information was provided by the survey firm to explain why 345 persons did not complete these questions. Some participants might have expressed they did not want to complete questions about child development or in some cases, the interviewer might have run out of time, as these questions were asked at the end of the survey, and perhaps not systematically asked. We added it as a limitation (lines 300-303): “Despite random sampling, selection bias might have occurred at different stages of the study, and we have no information about why some families having a child 0-5 years old did not respond to the questions about children’s development.

Point 3: Third, Table 3 reports mother and father differences but it is unclear if both the mother and father in each family were interviewed or if this just represents who happened to be interviewed in each family (with one interview per family).

Response 3: A clarification was added at the end of the sample section (section2.2, lines 101-102) “Either the father or the mother of this child could respond to these questions, depending who was completing the survey.”

Point 4: More significant, there seem to be two major deficits in the way data were collected that present large limitations on the usefulness of the study.  One is that there appears to be no data on how many children are in child care settings where presumably many of the activities asked about do occur. So were families asked only about home-based activities? If families had multiple children, they were asked to only describe the oldest child which means the sample might be skewed to older children who more likely would be in nursery or preschool settings?  If this isn't know are there other data about preschool enrollment/child care rates in the area that might lend some insight?

Response 4: For the point about childcare, we agree it could have been interesting to present data about child use of childcare setting. For your information, a question about childcare was part of our original survey questionnaire but modifications made to the survey, outside of our research team, resulted in unusable data. We added attendance at childcare centre as a data that would have been interesting to have in our discussion: “Several variables that may have being helpful for our analysis, such as child age, attendance at childcare center and parental education were not available. Further study should be conducted to explore effect of these variables on parental concerns and children participation in activities.”. (lines 296-297)

To acknowledge that questions related more to home-based activities, we added an example of question in the data collection section: “Survey questions about children’s activity participation inquired about reading activities (“During the past 12 months, how often did you and your child, or any other adult in the household with your child, do reading activities (eg reading a book or telling a story, going to the library or bookstore)?”)”. (lines 111-114)

Point 5: The age of the child is also extremely important since for parents of infants or toddlers, some of the activities might be less relevant or harder for them to answer accurately (for example would parents interpret 'tummy time' as physical exercise/motor development?)  These deficits in study design make the findings a lot less useful. Parental expectations and knowledge would also figure into their ratings and might differ by age of child, and if they had only one or more than one children. The latter analysis seemingly could be done. 

While these issues are alluded to in the discussion, along with limitations about no age data, the authors need to expand the discussion on these points and link implications or actions to a range of possible interpretations, including possible follow up studies that obtain more in depth information from a smaller sample of families with young children.

Response 5: We agree that the fact we had not access to children age is an important limitation and we acknowledged it in the limitations section: “Several variables that may have being helpful for our analysis, such as child age, attendance at childcare center and parental education were not available.” (lines 296-297). We emphasised the aspect of parental expectations for younger children by adding the example of tummy time in the following sentence : “Our results may, however, need to be interpreted with caution, as parents may not be aware of the activities considered as fine motor or physical activities, especially for children younger than 1 year (eg tummy time could have been considered or not as motor activities depending on parental perceptions, knowledge and expectations related to physical activity).”(lines 2016-217). Moreover, a sentence was added at lines 297-299 about follow up studies needed to explore the effect of children’s age, childcare attendance and parental education on parental concerns and children participation in activities: “Further study should be conducted to explore effect of these variables on parental concerns and children participation in activities.”

Thank you for considering this manuscript for publication in IJERPH.

Sincerely,

Gabrielle Pratte

On behalf of the study co-authors

Reviewer 3 Report

Dear authors!

Congratulations for your work.

I may only recommend to introduce some citations of theoretical and methodological references about psychological development and school readiness of pre-school children.

I also recommend to include all (or the most important) questions to the parents.

Thank you for your work

Author Response

Thank you very much for reviewing our manuscript. Modifications made as per your suggestions (bold-face type), are detailed below (in italic). Modification of the original text is highlighted in red in the text.

Point 1: I may only recommend to introduce some citations of theoretical and methodological references about psychological development and school readiness of pre-school children.

Response 1: Some details about theoretical choices related to the focus on “child global development” instead of “school readiness” were added in the introduction section : “In accordance with governmental guidelines, we choose a global development perspective to describe pre-schooler’s development, rather than a school readiness perspective, that is sometime understood as limited to cognitive and language skills or specific knowledge [16,17]. To do so, we used a framework increasingly used in Quebec, and among health professionals, including five developmental domains: affective (e.g. managing emotions, self-confidence), cognitive (e.g. learn, memorize), social (e.g. make friends), language (e.g. speak and understand) and physical and motor development (e.g. run, manipulate objects)[16].” (lines 67-73)

Point 2: I also recommend to include all (or the most important) questions to the parents.

Response 2:  Section 2.3 has been expanded to add more examples of survey questions (lines 111-119) : “Survey questions about children’s activity participation inquired about reading activities (“During the past 12 months, how often did you and your child, or any other adult in the household with your child, do reading activities (eg reading a book or telling a story, going to the library or bookstore)?”), fine motor activities (“During the past 12 months, how often did your child engage in fine motor skills such as tinkering, drawing, gluing or cutting, or activities such as performing arts, painting, sculpture?”), and physical activities (“During the past 12 months, how often did your child participate in physical activity or free sport with or without a coach or instructor (eg karate class, playing ball, jumping rope, riding a bike, go swimming)?”) with 7 response options varying from “Rarely or never” to “Daily”.

Thank you for considering this manuscript for publication in IJERPH.

Sincerely,

Gabrielle Pratte

On behalf of the study co-authors

Round 2

Reviewer 2 Report

To the extent possible, authors have clarified methods issues and added information to address sampling bias and study limitations.  Given the limitations of the survey utilized the manuscript presents adequate results and interpretation.  One minor point is that the results section does not give a complete picture of the data, reporting only the majority frequency (weekly) of parental developmental activities. This doesn't become clear until the discussion where the authors then report the percent of parents who indicated only monthly or less activities used as an illustrative point of concern and discuss the importance of daily reading activities. 

Author Response

Point 1 : To the extent possible, authors have clarified methods issues and added information to address sampling bias and study limitations.  Given the limitations of the survey utilized the manuscript presents adequate results and interpretation.  One minor point is that the results section does not give a complete picture of the data, reporting only the majority frequency (weekly) of parental developmental activities. This doesn't become clear until the discussion where the authors then report the percent of parents who indicated only monthly or less activities used as an illustrative point of concern and discuss the importance of daily reading activities. 

Response 1: Thank you for comment. We clarified the response options in the methods section (lines 118-120): "with 7 response options (“Rarely or never”, “Less than once a month”, “Once a month”, “A few times a month”, “Once per week”, “A few times a week”)" and the recategorization in the analysis section (lines 137-139) "To ease data reporting and comparison, data were clustered as appropriate (e.g. response options were dichotomized in ‘’at least once a week” and “less than once a week” for frequency of children’s participation in activities)". Modifications are in red in the text.